# Initial Validation of a Behavioral Phenotyping Model for Substance Use Disorder

**DOI:** 10.3390/ijerph21010014

**Published:** 2023-12-21

**Authors:** Lori Keyser-Marcus, Tatiana Ramey, James M. Bjork, Caitlin E. Martin, Roy Sabo, F. Gerard Moeller

**Affiliations:** 1Department of Psychiatry, Division of Addictions, Institute for Drug and Alcohol Studies, Virginia Commonwealth University, Richmond, VA 23219, USA; 2Division of Therapeutics and Medical Consequences, National Institute on Drug Abuse (NIDA), Gaithersburg, MD 20877, USA; 3Department of Obstetrics and Gynecology, Institute for Drug and Alcohol Studies, Virginia Commonwealth University, Richmond, VA 23219, USA; caitlin.martin@vcuhealth.org; 4Department of Biostatistics, Virginia Commonwealth University, Richmond, VA 23219, USA; roy.sabo@vcuhealth.org

**Keywords:** deep phenotyping, addiction, factor analysis, treatment matching

## Abstract

Standard nosological systems, such as DSM-5 or ICD-10, are relied upon as the diagnostic basis when developing treatments for individuals with substance use disorder (SUD). Unfortunately, the vast heterogeneity of individuals within a given SUD diagnosis results in a variable treatment response and/or difficulties ascertaining the efficacy signal in clinical trials of drug development. Emerging precision medicine methods focusing on targeted treatments based on phenotypic subtypes rather than diagnosis are being explored as alternatives. The goal of the present study was to provide initial validation of emergent subtypes identified by an addiction-focused phenotyping battery. Secondary data collected as part of a feasibility study of the NIDA phenotyping battery were utilized. Participants completed self-report measures and behavioral tasks across six neurofunctional domains. Exploratory and confirmatory factor analysis (EFA/CFA) were conducted. A three-factor model consisting of negative emotionality, attention/concentration, and interoception and mindfulness, as well as a four-factor model adding a second negative emotion domain, emerged from the EFA as candidate models. The CFA of these models did not result in a good fit, possibly resulting from small sample sizes that hindered statistical power.

## 1. Introduction

Drug addiction is a complex, chronic, relapsing brain disease. As such, treatment interventions for addiction have had variable success. For example, although both agonist and antagonist pharmacotherapies for opioid use disorder and alcohol use disorder are considered gold-standard treatment strategies, neither has overcome obstacles related to medication compliance and/or drug abstinence for many individuals. Further, attempts at developing pharmacotherapies for stimulant use disorders have failed to result in FDA-approved medications. Given the heterogeneity in participants’ symptomatology among persons within a particular SUD diagnosis, a “one size fits all” approach to treatment based solely on DSM diagnosis is insufficient. While DSM-based classification methods have been somewhat effective for the identification and broad-based treatment of individuals with SUD, the resulting limited sensitivity and specificity of deriving a diagnosis based on the consequences of substance use remains a concern [1]. Consequently, the field has called for clinically meaningful subtyping of persons with a specific SUD, such as based on emergent clusters of differing neurobehavioral endophenotypes [2].

General psychiatry has been similarly fraught with unpredictability in treatment efficacy across (and within) mental health conditions, as many patient factors and behavioral characteristics beyond presenting diagnosis contribute to the level of treatment effectiveness of a given (pharmacological and/or psychosocial) intervention. Heavy reliance on DSM classification as a tool to enroll in clinical trials and advance and tailor treatment for patients with a given psychiatric condition has fallen short. Due to the vast permutations of endorsed symptoms, there is broad heterogeneity within a DSM diagnostic category, including all comorbid diagnoses, resulting in variable patient outcomes. The advent of the NIMH Research Domain Criteria (RDoC), a clinical neuroscience-based framework, has shifted the focus of identifying new targets for the treatment of mental disorders solely based on presenting symptoms within the DSM system to a model that focuses on individual neurocircuit function at different levels of analysis and incorporates more information regarding the etiologic and pathophysiologic mechanisms of disordered behavior, utilizing neuroimaging and genetic data [3]. This type of framework has since been adopted for research purposes by the National Institute on Alcoholism and Alcohol Abuse (NIAAA) for Alcohol Use Disorders, termed the Alcohol Addiction Research Domain Criteria (AARDoC) [4].

To probe for different potential subtypes of mental disorders, complementary clinical assessment-based frameworks have emerged, including the NIAAA Addictions Neuroclinical Assessment (ANA), a deep phenotyping assessment battery that deploys assessments that ostensibly capture three core neurofunctional domains relevant to the development and maintenance of addiction: incentive salience, negative emotionality, and executive function (EF) [5,6]. These three domains emerged largely from preclinical models and were extended into the clinical realm. These more detailed assessments, such as the ANA, can allow for more targeted intervention efforts using a precision medicine approach, where a certain medication could be matched to a patient with a pronounced deficit in a certain domain. Subsequently, the NIDA Phenotyping Assessment Battery (PhAB) expanded the ANA model to include three additional domains: Interoception, Metacognition, and Sleep/Circadian Rhythm. A recent feasibility study of the PhAB confirmed that the battery could be completed with minimal participant burden (averaging approximately 1.5 h for phenotyping assessments and up to 1.5 h for the broad, complementary platform measures that are flexible in terms of supplementing the PhAB and can be adjusted based on the need). Pilot PhAB administration had demonstrated high rates of participant satisfaction and minimal cost time/financial cost to investigators [7].

A challenge with identifying phenotypic subtypes of SUD, however, is the intercorrelation between individual metrics of impaired cognitive control and impaired emotion control [8,9]. These interrelationships may collectively stem from how factor analyses of the structure of psychopathology itself indicate that a host of aberrations can be captured as shared variance in an omnibus “p” (or psychopathology) factor [10]. These intercorrelations or p-factor findings can also be genetically accounted for in that polygenic risk for substance use frequency (e.g., alcohol use quantity) is distinct from genetic risk markers for *disordered us.* The latter of which genetically correlate with the risk of other non-SUD mental disorders, generally [11]. Clinically, these intercorrelated impairments in cognitive and emotional control can in turn contribute to assessment findings of impaired sleep, lower quality of life, and reduced daily function.

The goal of the present study was to determine whether the host of individual SUD-relevant phenotypes captured in the theoretical 6-domain PhAB model aggregate into respective emergent domains and whether the symptom/phenotypic clusters that emerge reflect the assumed ANA domains. We factor-analyzed the phenotyping assessment data collected during the feasibility trial to test the expanded PhAB six-domain theoretical model using techniques similar to Kwako et al. [12]. We hypothesized that the factor analysis of the PhAB metrics would segregate into mechanistically meaningful and distinct phenotypic domains.

## 2. Materials and Methods

### 2.1. Study Participants

Participants for the present study were recruited as part of a larger feasibility study of a Phenotyping Assessment Battery (PhAB) developed by NIDA, described in detail elsewhere [7]. Individuals with one or more primary SUD diagnoses (Opioid, Cocaine, and Cannabis) were recruited, as well as a comparison group without SUD. Eligibility criteria for the larger study intentionally allowed for the recruitment of a “real-world” SUD sample with heterogeneous comorbidities, such that individuals with comorbid SUD diagnoses were considered eligible, although severe Alcohol Use Disorder was exclusionary. This study was reviewed and approved by the Virginia Commonwealth University Institutional Review Board. All participants provided written informed consent prior to participation.

### 2.2. Phenotypic Assessment Battery

Participants completed a screening visit to determine eligibility, followed by a phenotyping visit, which lasted approximately 3.5 to 4 h, on average (including rest breaks). The phenotyping visit included both the (core) phenotyping measures as well as a supplemental “platform” assessment battery composed of self-reported symptom severity scales (e.g., Visual Analog Scale for Pain (VAS-Pain)), substance use measures (e.g., Timeline Follow-back), and IQ/performance measures. The Phenotyping measures are listed by (intended) domain in Table 1.

### 2.3. Selection of Measures for Data Analyses

Five of the phenotyping battery measures were eliminated from the present analyses for various reasons. The Attentional Network Task (ANT) could not be used, as the version of the measure was changed after interim analyses noted that the time required to complete the ANT initially selected for use was too long (approximately 20 min) to support its feasibility for future use. The Hypothetical Purchase Task (a behavioral measure of valuation associated with primary substance use) was also not included, as the measure was not administered to non-drug-using control participants, and individuals with SUD who were in remission experienced difficulties assigning values to the items. Data from the line counting/cue interference task were eliminated from consideration as the data were substance-specific and only available for participants with opioid use disorder. Lastly, the Emotional go/Nogo and Stop Signal tasks were excluded due to behavioral evidence that a sizable number of participants did not understand or comply with task instructions.

The following assessments remain for consideration in the factor analysis within each of the six PhAB neurofunctional domains: Cognition domain: backward visual digit span (bML, bMS, and bTE TT scores); 5 Trial delay discounting (approximated discounting constant); Reward domain: SUPPS-P impulsive behavior scale (NURG, PRUG, PERS, PREM, and SS subscales); Interoception domain: Multidimensional Assessment of Interoceptive Awareness (MAIA: noticing, not distracting, not worrying, attention regulation, emotional awareness, self-regulation, body listening, and trusting subscales); Negative emotionality: Distress tolerance scale (tolerance, absorption, appraisal, regulation, and DTS-G (global)), PROMIS depression (total score), PROMIS anxiety (total score), Buss-Perry scale (physical aggression, verbal aggression, anger, and hostility subscales), Snaith Hamilton Pleasure Scale (SHAPS: total score); Sleep domain: Pittsburgh Sleep Quality Index (PSQI: total score); Metacognition domain: the Metacognitions Questionnaire (MCQ-30: general metacognition, positive beliefs about worry, negative beliefs about uncontrollability and danger of worry, cognitive confidence, need for control, and cognitive self-consciousness subscales).

### 2.4. Statistical Analysis

Exploratory and confirmatory factor analyses were conducted using the approach described by Kwako et al. [12]. The data were randomly split into two halves (*n* = 146; *n* = 145) for training and testing. Exploratory factor analysis (EFA) were performed on the training data to identify latent factors underlying the indicator variables. Weighted least square with full-information maximum likelihood along with *oblimin* rotation was used for estimation, as some measures showed departures from normality. Fit indices included the root mean square error of approximation (RMSEA) with a 95% confidence interval, the comparative fit index (CFI), and the Tucker-Lewis Index (TLI). Based on commonly accepted thresholds, a model with good fit will have RMSEA below 0.06 and both CFI and TLI above 0.95. Indicators were removed from consideration due to multicollinearity with other indicators in their domains and to improve fit metrics, with removal based on a low coefficient of determination between an indicator and the drug-use classification. The CFA were performed on the testing data with any factor loadings less than 0.35 set to 0; modification indices were explored and applied if model fit was improved. Patient sex and race were compared between groups using chi-square tests, while age was compared between groups using analysis of variance. Analyses were conducted using SAS statistical software (version 9.4, Cary, NC, USA), with the CALIS procedure.

## 3. Results

### 3.1. Sample

The present sample included 287 individuals, with 143 (50%) having primary cocaine and/or opioid use disorders. Approximately half (48%) of the sample were female, with greater representation among healthy controls (63%) than in either of the SUD groups (*p*-value < 0.0033). Almost two-thirds (64%) identified as Black/African American, accounting for larger percentages than Whites with Cocaine or Opioid use disorders (83% vs. 15%) or with Cannabis user disorder (56% vs. 38%), with similar numbers of Health Controls (40% vs. 44%). The average age of participants with cocaine and/or opioid use disorder was approximately 10 years older (45.9 years) than the healthy controls (35.9 years, *p*-value < 0.0001) and individuals with cannabis use disorder (34.8 years, *p*-value < 0.0001). Participant demographic characteristics and indicator measures are summarized in Table 2.

### 3.2. Factor Analysis

Two models were identified in the exploratory factor analysis: a three-factor model (RMSEA = 0.08, CFI = 0.99, TLI = 0.99) and a four-factor model (RMSEA = 0.05, CFI = 0.99, TLI = 0.99); both the one-factor (RMSEA = 0.26, CFI = 0.99, TLI = 0.89) and two-factor (RMSEA = 0.13, CFI = 0.99, TLI = 0.97) models had unacceptable fit criteria, so these alternatives were not considered further. Factor loadings from the three-factor model are reported in Table 3. The first factor that emerged was termed Negative Emotionality, due to endorsements of a tendency to act rashly under negative emotions, to act without thinking, to feel anxious and depressed, and to act physically aggressive. This factor consisted of positive loadings across the PhAB reward, negative emotionality, metacognition, and sleep domains: the SUPPS-S (negative urgency (N-URG (0.84)) and premeditation (PREM (0.486) subscales)), PROMIS Anxiety (0.866) and PROMIS Depression (0.775) total scores, the MCQ-30 (negative beliefs about uncontrollability and danger of worry subscale (0.845)), the PSQI total score (0.602), SHAPS (anhedonia) total score (0.396), and Buss Perry Physical Aggression subscale (0.537). Additionally, a negative loading (−0.685) on the DTS Tolerance subscale was also noted for this factor. The second factor identified, Attention and Concentration, featured high positive loadings on both the bMS and bTE-ML Backward Visual Digit Span measures (0.934 and 0.955, respectively). The third factor identified, Interoception and Mindfulness, was distinguished by high positive loadings in the interoception domain (MAIA-Noticing (0.605) and MAIA-Trusting (0.745) subscales) and a negative loading (−0.511) in the reward domain (SUPPS-P PREM (lack of premeditation)). This factor is characterized by awareness of body sensations and experiencing one’s body as safe and trustworthy, as well as planfulness and thinking before acting.

Factor loadings from the four-factor model are reported in Table 4. The first three factors resemble those from the three-factor model, with two slight differences. The first factor (Negative Emotionality) has similar positive and negative loadings between models, except the positive loadings for the Buss Perry Physical Aggression subscale and the SHAPS total score have been removed and added to the new fourth factor (with loadings of 0.754 and 0.466, respectively), which also has a positive loading for negative urgency (0.528). The second factor (Attention and Concentration) was almost identical to that from the three-factor model, as was the third factor (Interoception and Mindfulness), which added a positive factor loading (0.367) from the cognition domain (5 Trial Delay Discounting Total).

Confirmatory factor analysis in the testing data set, however, did not result in acceptable fit for the three-factor model (RMSEA = 0.16, CFI = 0.99, TLI = 0.97) nor the four-factor model (RMSEA = 0.13, CFI = 0.99, TLI = 0.98). Confirmatory factor analysis in the combined data set also did not meet all fit criteria in the three-factor model (RMSEA = 0.06, CFI = 0.99, TLI = 0.91), but did meet all criteria in the four-factor model (RMSEA = 0.04, CFI = 0.99, TLI = 0.97).

## 4. Discussion

Factor analysis of the PhAB was performed to determine whether latent phenotypic factors derived from the EFA and CFA emerged that were consistent with the conceptualized subdomains of SUD symptomatology, akin to the emergent factors from the ANA-specific assessment that supported the ANA [12]. Results from the present study provide preliminary (albeit tenuous) corroboration of a neurobiology-based core comprised of a parallel 3-domain model related to that noted for AUD [5]. Although the model was not confirmed, the present 3-domain and 4-domain models derived from a sample of individuals with opioid, cocaine, and cannabis use disorders lend support to further exploring the utility of the deep phenotyping method as a viable means to determine treatment targets beyond the dichotomous focus on abstinence with a more precision medicine-based approach. Despite the a priori intention that the PhAB battery will expand the ANA, including six domains of symptomatology, findings from the factor analysis of measures included in the PhAB six-domain model of addiction resulted in a three-factor model, which included Negative Emotionality, Attention and Concentration, and Interoception and Mindfulness, and a four-factor model that added a second Negative Emotion domain.

Only one of the three addiction domains identified by the ANA and also hypothesized in the NIDA PhAB theoretical model, Negative Emotionality, was revealed in the current factor structure. This factor included loadings from the measures included in the PhAB reward (negative urgency and lack of premeditation), Metacognition (negative metacognitive beliefs), Negative Emotion (anxiety and distress tolerance), and Sleep domains. One of the greatest contributors to this factor was a metacognition variable (MCQ-30 subscale negative beliefs about the uncontrollability of dangerous thoughts). Metacognition could be defined as “the psychological structures, knowledge, events, and processes that are involved in the control, modification, and interpretation of thinking itself” [13]. This heavy factor loading is not surprising given the growing body of literature supporting the vital role of metacognition in both establishing and maintaining addictive behavior [14,15], including the relationship between negative metacognitive beliefs and substance craving and use [14,16]. 

Negative metacognitions are thought to maintain substance misuse due to a bidirectional relationship that exists between negative affect (anxiety, depression) and negative metacognitive beliefs (e.g., I can’t control my cocaine use), which leads to a loss of behavioral control/failure to implement effective coping strategies and reinforces ruminative thinking and subsequent negative emotions [14]. Subsequently, the contribution of anxiety to the Negative Emotionality factor is also not surprising. Further, this factor continues to support the strength of the association between impulsivity and SUD, as evidenced by the contribution of the negative urgency subscale (failure to inhibit behavioral responses in the context of negative events/emotions/triggers), as well as general inhibitory control deficits from the lack of premeditation subscale.

The presence of physical aggression in the Negative Emotionality cluster is not surprising in light of its well-documented and complex incidence in the SUD population generally, coupled with the propensity for reactive aggression to stem from a negative mood (e.g., [17]). For example, research has demonstrated strong associations between personal characteristics of aggression, impulsivity, antisocial personality disorder, and SUD [18], as well as direct correlations between substance use and aggressive behavior [19]. The negative loading of distress tolerance (DT) on this factor may explain increased aggressiveness. The relationship between distress tolerance and SUD is an inverse one, as supported by the literature and by clinical lore, with individuals endorsing lower levels of DT exhibiting greater vulnerability to developing a SUD, suffering more negative consequences associated with substance use, and having poorer treatment outcomes [20,21,22]. However, increased aggression in SUD is not universal and may even be more substance-specific [23] relative to more ubiquitous findings of increased negative emotionality as a cause or consequence of SUD.

Lastly, the contributions of depression, anxiety, and sleep to the Negative Emotionality factor are also not surprising. We suspect that this negative emotionality factor, detectable in both our analysis and the ANA analysis, may reflect the overall p factor found in bifactor modeling of the structure of psychiatric symptomatology generally [24], in that negative emotions and beliefs are a cross-cutting feature of several mental disorders. Sleep dysfunction is common across mental illnesses [25], but may be exacerbated by the neurotoxic and acute withdrawal effects of drugs of abuse [26]. Sleep disturbance is also a common complaint among individuals with SUD and may exacerbate other addictions and/or psychiatric sequelae and compromise SUD treatment outcomes [27,28,29,30,31]. Notably, sleep disturbance is closely linked to heightened pain sensitivity, impulsivity, mood instability, and stress in various clinical populations [32,33], especially among patients with SUD. Sleep disturbance is gaining attention as a potential treatment target for interventions to address not only sleep disturbance itself but also other SUD domains within a treatment context [31]. Our findings further support this emerging evidence, highlighting how targeted sleep disturbance among SUD patients may be an effective avenue to simultaneously address multiple underlying mechanisms of SUD, specifically those reflected in our Negative Emotionality phenotype. Taken together, the loadings that contribute to the Negative Emotionality factor appear to form a composite, with empirical support noting dysfunction in one of the respective areas contributing to problems in one or more related areas and each of the loadings demonstrating associations with SUD outcomes.

Attention and Concentration were the next factors to emerge, consisting of two loadings from the Backward Visual Digit Span. Surprisingly, none of the other PhAB cognition-related measures included in the model contributed to any factors in the current analyses. One might argue that EF in the current model was also represented by the precognition constructs of metacognition and interoception, which loaded onto each of the other factors. A study performed by Kraft and colleagues examined the relationship between dysfunctional metacognitive beliefs and executive control [34]. They found that individuals who endorsed negative metacognitive beliefs (about uncontrollability and danger of worry) experienced greater difficulties in shifting between mental sets on a neuropsychological task. This reflects the cognitive rigidity and other EF deficits found in ruminative disorders more broadly [35] and suggests that the cognitive load associated with negative metacognitions may inhibit EF. In their review, Verdejo-Garcia and colleagues [36] posit that individual differences with regard to each of interceptive sensitivity, the accuracy with which sensations are perceived, and the appraisal of the sensation may account for subsequent heterogeneity among individuals within a given SUD diagnosis in that all contribute to the cognitive-affective processing of stimuli, but with different underpinnings or manifestations.

The final factor identified, Interoception and Mindfulness, appears to describe a pattern in which somatic sensations experienced are perceived and interpreted in a manner that leads to an adaptive rather than maladaptive (e.g., impulsive, anxious) response. The two highest loadings onto this factor were the Noticing subscale (e.g., awareness of body sensations as uncomfortable, comfortable, or neutral) and the Trusting subscale (e.g., experiencing one’s body as safe and trustworthy) from the Multidimensional Assessment of Interoceptive Awareness (MAIA). Interoception as a construct can also be defined in terms of improved bodily awareness, such as an individual’s awareness of somatic sensations, ranging from subtle observations such as feeling one’s own heartbeat to more potent ones such as hunger, thirst, cold, etc. The negative loading for the Lack of Premeditation subscale of the SUPPS-P complements the MAIA loadings, as it indicates a mindful approach to assessing events prior to responding. The primary responsibility of interoception appears to be that of a homeostatic regulator, which compels an individual to assess and act on the information provided to maintain internal equilibrium. The insula has been identified as central to interoceptive processing, and interoceptive irregularities have been implicated in a number of psychiatric disorders, including SUD [36,37,38,39].

## 5. Conclusions

The emergent factors suggest that persons with SUD may differ on dimensions of Negative Emotionality, Interoception, Attention, and Concentration, each of which may be targeted with different therapeutic treatment approaches, such as mindfulness-based interventions, cognitive behavioral therapy, etc. Consequently, rather than using a “one-size fits all” approach, identifying neurobehavioral features that distinguish between individuals would allow for tailored treatment planning. Future directions may include the exploration of the outer boundaries of social cognition, e.g., how metacognitive beliefs and Theory of Mind (ToM) factors are shaped by drug use and also represent targets for treatment intervention to positively affect not only the individual but also indirectly frame a positive change in the proximal ecosystem that surrounds the individual with addiction (e.g., family). Another direction would be to delve deeper into the intermediate endophenotype and attempt deep phenotyping to arrive at risk and/or treatment outcome-predictive neurofunctional patient profiles, which would have implications for prevention as well as advancing targeted new treatments for SUD.

Sex (a biological variable) and gender (a continuum of sociocultural constructs)-related factors, as seen in other chronic medical conditions [40], are well-known variables that modify addiction and recovery trajectories [41]. Sex-based characterization of neurofunctional subtypes in people with SUD is presently lacking, yet this knowledge is essential to establishing a foundation for the development of targeted interventions to achieve sex and gender equity in SUD clinical care. Thus, future phenotyping work on addictions, especially those in larger samples, should intentionally prioritize sex and gender-specific analyses.

The primary weakness of the current study is the relatively small sample size used in the factor analysis (*N* = 287; *n* = 146 for EFA and *n* = 145 for CFA), which mostly limited the number of total variables we could include in the analyses and necessitated selecting the most promising variables before fitting the EFA (which still greatly exceeded the recommended cases to variable ratio and most likely contributed to the ill-fitting model in the CFA). Moreover, complete factor analysis of the PhAB battery as intended were precluded by the failure to obtain valid data from a substantial number of participants due to failures to understand (or comply with) experimenter instructions. While this instance of omitted data are crude but indirect evidence of cognitive impairment in SUD itself, the reduced number of valid data points would have required truncation of the entire analytic sample, in that our analytic approach did not allow for missing or interpolated data. A larger sample, coupled with the universal inclusion of valid data from simpler or briefer cognitive performance tasks, may have yielded a factor solution more akin to the symptom domains that were configured a priori. However, this filtering could potentially reduce the possibility of overfitting and over predicting in both the EFA and CFA models. Finally, we note that the self-report trait-like metrics of cognition that we did retain may have more test-retest reliability than computerized neurocognitive performance measures [42], where, for example, self-report versus laboratory task-derived metrics of impulsivity typically correlate poorly within-subject [43].

## Figures and Tables

**Table 1 ijerph-21-00014-t001:** PhAB Phenotyping Measures.

PhAB Domain	Measure
Cognition	Backwards Visual Digit Span
	Attentional Network Task (ANT) *
	5 Trial Delay Discounting
	Stop Signal Reaction Task *
Reward	Hypothetical Purchase Task *
	Line Counting/Cue Interference Task *
	SUPPS-P Impulsive Behavior Scale
Interoception	Multidimensional Assessment of Interoceptive Awareness (MAIA)
Negative Emotionality	Emotional Go/Nogo Task *
	Distress Tolerance
	PROMIS- Depression
	PROMIS-Anxiety
	Buss Perry Aggression Scale
	Snaith-Hamilton Pleasure Scale
Metacognition	Metacognition Questionnaire (MCQ-30)
Sleep	Pittsburgh Sleep Quality Index-Revised (PSQI)

Note: Measures marked with an “*” were not included in the factor analysis.

**Table 2 ijerph-21-00014-t002:** Demographic characteristics and indicator measures by primary SUD diagnosis.

	Healthy Controls (*n* = 96)		Cocaine or Opioid (*n* = 143)		Cannabis (*n* = 48)	
**Measures**	** *n* **	**%**	** *n* **	**%**	** *n* **	**%**
**Female**	60	63%	59	41%	20	42%
**White**	42	44%	21	15%	18	38%
	**Mean**	**SD**	**Mean**	**SD**	**Mean**	**SD**
**Age**	35.9	14.9	45.9	11.4	34.8	13.4
**Snaith Hamilton Pleasure Scale (Total)**	0.6	1.1	1.9	3.2	1.7	3.0
**5 Trial Delay Discounting (Total)**	1.9	5.9	3.6	6.9	2.6	5.9
**PROMIS Depression Total**	8.6	3.3	9.1	3.4	9.7	4.1
**PROMIS Anxiety Total**	7.5	3.3	8.5	3.5	8.4	3.7
**Pittsburgh Sleep Quality Index (PSQI)**	5.4	2.8	7.1	3.6	7.3	3.1
**Backward Visual Digit Span**						
Max backward digits recalled (bML)	6.3	1.6	5.9	2.1	6.1	1.8
Digit span expended correctly at 50% (bMS)	5.7	1.6	5.3	2.1	5.6	1.8
Two-error max length (bTE ML)	5.5	1.7	5.0	2.2	5.3	1.8
Two-error total trials (bTE TT)	5.1	2.0	4.7	2.6	4.9	2.2
**SUPPS-P Impulsive Behavior Scale**						
Negative Urgency (N-URG)	7.2	2.6	9.0	2.8	8.9	3.1
Positive Urgency (P-URG)	6.4	2.2	7.5	2.9	7.6	2.8
Lack of Perseverance (PERS)	6.7	1.8	6.4	2.1	7.0	2.6
Lack of Premeditation (PREM)	6.5	2.1	6.9	2.5	6.7	2.2
Sensation Seeking (SS)	9.8	3.0	9.2	2.9	10.4	3.2
**MAIA Interoception Scale**						
Noticing	3.4	1.1	3.4	1.2	3.4	1.1
Not Distracting	2.4	1.1	2.4	1.2	2.3	1.3
Not Worrying	3.2	1.0	2.7	1.1	2.9	1.0
Attention Regulation	3.2	0.9	3.2	1.1	3.4	1.0
Emotional Awareness	3.3	1.0	3.4	1.2	3.8	0.9
Self Regulation	3.1	1.2	3.2	1.1	3.5	0.9
Body Listening	2.3	1.5	2.4	1.4	2.5	1.2
Trusting	3.5	1.3	3.5	1.3	3.8	1.0
**Distress Tolerance Scale (DTS)**						
Total Score (DTS-G)	3.7	0.7	3.5	0.8	3.5	0.8
Tolerance	3.8	0.8	3.5	1.0	3.5	0.9
Absorption	3.7	1.0	3.6	1.0	3.5	1.0
Appraisal	3.9	0.8	3.6	0.8	3.6	0.9
Regulation	3.6	1.0	3.2	1.1	3.4	0.9
**Buss Perry Aggression Scale**						
Physical Aggression	16.5	5.3	23.0	6.9	22.7	7.1
Verbal Aggression	13.5	4.3	14.6	3.6	14.8	4.1
Anger	14.1	5.0	16.7	5.4	17.1	6.3
Hostility	19.5	6.7	21.8	6.7	21.2	7.9
**Metacognition Questionnaire (MCQ-30)**						
General Metacognition	56.5	13.2	57.0	11.2	61.1	13.6
Positive Beliefs about Worry	0.4	0.3	0.4	0.3	0.4	0.3
Negative Beliefs: Uncontrollable Danger	0.2	1.1	0.5	1.0	0.5	1.1
Cognitive Confidence	0.0	0.8	0.1	0.9	0.3	1.1
Need for Control	1.1	1.3	1.5	1.3	1.8	1.5
Cognitive Self-Consciousness	1.0	1.0	0.8	0.8	1.2	0.9

**Table 3 ijerph-21-00014-t003:** Factor loadings from exploratory 3-factor analysis.

Domain	Indicator	Negative Emotionality	Attention and Concentration	Interoception and Mindfulness
Cognition	bMS (Backward Visual Digit Span)	−0.041	**0.934**	−0.062
	bTE-ML (Backward Visual Digit Span)	−0.078	**0.955**	0.039
	5 Trial Delay Discounting Total	0.121	0.106	0.313
Reward	Negative urgency (N-URG; SUPPs-P)	**0.840**	0.168	−0.166
	Lack of Premeditation (PREM; SUPPs-P)	**0.486**	0.072	**−0.511**
Intercoception	Noticing (MAIA Interoception Scale)	0.124	−0.096	**0.605**
	Trusting (MAIA Interoception Scale)	−0.238	−0.024	**0.745**
Negative Emotionality	Distress Tolerance Scale Total	**−0.685**	0.091	−0.094
	PROMIS Depression Total	**0.775**	−0.007	−0.029
	PROMIS Anxiety Total	**0.866**	−0.111	0.034
	Physical_Aggression_(Buss_Perry Aggression Scale)	**0.537**	0.153	0.196
	Snaith Hamilton Pleasure Scale Total	**0.396**	−0.092	−0.034
Metacognition	Negative Beliefs: Uncontrollable Danger (Metacognition Questionnaire)	**0.845**	0.035	0.199
Sleep	Pittsburgh Sleep Quality Index Total	**0.602**	−0.028	−0.177

Bold type indicates the variable was a significant contributor to its corresponding factor.

**Table 4 ijerph-21-00014-t004:** Factor loadings from exploratory 4-factor analysis.

Domain	Indicator	Negative Emotionality	Attention and Concentration	Interoception and Mindfulness	Factor 4
Cognition	bMS (Backward Visugal Digit Span)	0.018	**0.961**	−0.023	−0.001
	bTE-ML_(Backward_Visual Digit Span)	−0.059	**0.940**	0.024	0.015
	5_Trial_Delay_Discounting Total	0.013	−0.042	**0.367**	0.156
Reward	Negative urgency (N-URG; SUPPs-P)	**0.507**	0.075	−0.132	**0.528**
	Lack of Premeditation (PREM; SUPPs-P)	**0.449**	0.087	**−0.507**	0.021
Intercoception	Noticing (MAIA Interoception Scale)	0.203	0.008	**0.680**	−0.090
	Trusting (MAIA Interoception Scale)	−0.103	0.079	**0.710**	−0.128
Negative Emotionality	Distress Tolerance Scale Total	**−0.675**	0.117	−0.129	−0.030
	PROMIS Depression Total	**0.821**	−0.084	−0.053	−0.062
	PROMIS Anxiety Total	**0.897**	−0.057	0.033	−0.008
	Physical Aggression (Buss Perry Aggression Scale)	−0.033	0.075	0.138	**0.754**
	Snaith Hamilton Pleasure Scale Total	0.038	−0.103	−0.125	**0.466**
Metacognition	Negative Beliefs: Uncontrollable Danger_Metacognition Questionnaire)	**0.815**	0.130	0.198	0.059
Sleep	Pittsburgh Sleep Quality Index Total	**0.545**	0.071	−0.196	0.087

Bold type indicates the variable was a significant contributor to its corresponding factor.

## Data Availability

The data presented in this study are available on request from the corresponding author.

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
