# Peer review of "Initial Validation of a Behavioral Phenotyping Model for Substance Use Disorder"

_ijerph, 2023, doi:10.3390/ijerph21010014_

Round 1

Reviewer 1 Report

Comments and Suggestions for Authors

Initial validation of a behavioral phenotyping model for substance use disorder 

This study aimed to provide initial validation of emergent phenotype subtypes identified by an addiction-focused phenotyping battery. The authors conducted Exploratory and confirmatory factor analysis with secondary data collected as part of a feasibility study of the NIDA Phenotyping battery. A three-factor model, emerged from the EFA, but the CFA did not provide good fit parameters. The authors explained that sample size limited statistical power. These factors limit the scope of this study. Besides, (and probably due to sample size problems in practice) some of the authors' choices in analysis strategy and presentation of results are not obvious to the reader, as explained below.

Comments:

Method: One of the study's arguments is “the heterogeneity of patients within a given SUD diagnosis” (abstract). However, if the goal of phenotyping is more precision, what's the interest of grouping together patients with cocaine use disorder and those with heroin use disorder in this analysis strategy?

2.1: 

·       how is primary SUD diagnose determined when there is comorbidity? (e.g., patients with both cocaine and cannabis UD)

·       Why severe alcohol use disorder was exclusionary?

Analysis and Table 2: 

·       Comparison group is presented as age-matched (2.1), however in Table 2 cocaine/opioid group is 10 years older than controls. How was age-matching verified?

·       Besides, no difference test is presented in table 2

·       For “primary SUD diagnosis”, why put cocaine and opiates together in the same group (and cannabis apart?)

3.2 and 4 : Discussion’s first paragraph (and the abstract) presents three domains in the model description: Negative Emotionality, Attention and Concentration, and Interoception and Mindfulness. However, in 3.2, only Negative Emotionality and Attention and Concentration are in text. Where is Interoception and Mindfulness?

Author Response

R1.1:  Method: One of the study's arguments is “the heterogeneity of patients within a given SUD diagnosis” (abstract). However, if the goal of phenotyping is more precision, what's the interest of grouping together patients with cocaine use disorder and those with heroin use disorder in this analysis strategy?

Response: Currently, DSM SUD diagnosis is one of the only means of identifying/categorizing individuals with SUD.  Subsequently, for recruitment purposes, we used DSM diagnoses for study eligibility and descriptive purposes.  Due to the high comorbidity of OUD and CocUD among our sample, these diagnoses were grouped together.  However, the factor analysis did not differentiate between participant subgroups. 

R1.2:   How is primary SUD diagnose determined when there is comorbidity? (e.g., patients with both cocaine and cannabis UD)

Response: The principal criteria used to determine primary SUD diagnosis was based on severity (using DSM criteria).  Additionally, one of measures includes a self report item re: preferred substance.  Due to the high comorbidity of OUD and CocUD, those diagnoses were merged into a single group for descriptive purposes.  Individuals in the CanUD group did not meet criteria for OUD or CocUD. 

R1.3:  Why severe alcohol use disorder was exclusionary?

Response: The eligibility criteria for the parent study from which the data were obtained restricted inclusion criteria to individuals with Opioid Use Disorder, Cocaine Use Disorder, and Cannabis Use Disorder, but made the exception for Alcohol Use Disorder (due to high rate of alcohol use among this population), with the caveat that individuals could not meet criteria for severe alcohol use disorder. 

R1.4:  Comparison group is presented as age-matched (2.1), however in Table 2 cocaine/opioid group is 10 years older than controls. How was age-matching verified?

Response:  Although our control group was recruited using many of the same inclusion and exclusion criteria as our SUD samples (including age range); we were mistaken in describing our sampling process as “matched.” Thus, natural variation in sampling lead to a younger control group. We have changed the sampling description in Section 2.1 to better-reflect our approach, and we deeply apologize for the confusion caused by our initial phrasing.

R1.5: Besides, no difference test is presented in table 2

Response:  To keep this large table from getting too complicated, we have added more description of these comparisons (as well as p-values from chi-square tests and analysis of variance) to the first paragraph of Section 3.

R1.6:   For “primary SUD diagnosis”, why put cocaine and opiates together in the same group (and cannabis apart?)

Response:  As noted above, cocaine and opiates were combined due to high comorbidity in those individuals.  However, the individuals in the Cannabis use disorder group, did not meet criteria for opioid or cocaine use disorder. 

R1.7:  Discussion’s first paragraph (and the abstract) presents three domains in the model description: Negative Emotionality, Attention and Concentration, and Interoception and Mindfulness. However, in 3.2, only Negative Emotionality and Attention and Concentration are in text. Where is Interoception and Mindfulness?

Response:  Thank you for drawing our attention to that oversight.  This must have occurred when transferring the manuscript onto the template.  This has now been updated to include Interoception and Mindfulness. 

Reviewer 2 Report

Comments and Suggestions for Authors

The primary objective of this manuscript is to undertake a comprehensive examination of the structural validity of a behavioral phenotyping model that pertains specifically to substance use disorder. The findings and subsequent reporting of this study may be of interest to researchers and healthcare practitioners. The introduction is well composed and substantiated.

My major criticism is the following:

- At the end of the introduction, I think it would be worth briefly summarizing what kind of structural arrangement we expect and what the hypotheses are.

- The description of the sample should be placed in the Study Participants section rather than the Results section.

- For the measures used (especially for the cognitive tasks), it would be important to examine the distributions (e.g., skewness, kurtosis), as this may determine the method of factor analysis.

- The age means for the three groups differ significantly. How this was addressed in the comparisons?

- When determining the dimensions of factor analysis, it would be worthwhile to use traditional indicators and procedures, e.g., KMO and parallel analysis, to determine the number of dimensions.

- For CFA, it would also be worthwhile to present other model fits (1, 2, or 4 factor or bifactor model), perhaps indicating the extent of residuals based on modification indices.

- I feel a problem is that self-report and cognitive outcomes are different types of data, and for example, some scales (e.g., visual digit span) are latent dimensions due to strong covariation, i.e., the ordering is primarily a methodological effect.

- Although the discourse is based on sound professional foundations, I believe that the resulting analysis and results need more care and deeper interpretation.

The data analysis does not seem convincing to me.

Author Response

R2.1: - At the end of the introduction, I think it would be worth briefly summarizing what kind of structural arrangement we expect and what the hypotheses are.

Response:  The hypotheses were expanded upon in the current version for clarification.

R2.2 The description of the sample should be placed in the Study Participants section rather than the Results section.

Response:  The initial description of study participants is noted in Section 2.1 (Study Participants). 

R2.3. For the measures used (especially for the cognitive tasks), it would be important to examine the distributions (e.g., skewness, kurtosis), as this may determine the method of factor analysis.

Response: Some of the distributions showed departures from normality, which is why we used robust weighted least squares (Muthén LK, Muthén BO: Mplus User’s Guide, 7th ed. Los Angeles, Muthén & Muthén, 2015). We have now added a declarative statement in Section 2.4 as the reason for using WLS.

R2.4.  The age means for the three groups differ significantly. How this was addressed in the comparisons?

Response:  Our control group was recruited using the many of the same inclusion and exclusion criteria as our SUD samples (including age range); we were mistaken in describing our sampling process as “matched.” Thus, natural variation in sampling lead to a younger control group. We have changed the sampling description in Section 2.1 to better-reflect our approach, and we deeply apologize for the confusion caused by our initial phrasing.

R2.5.  When determining the dimensions of factor analysis, it would be worthwhile to use traditional indicators and procedures, e.g., KMO and parallel analysis, to determine the number of dimensions.

Response:  While agree that these are viable (and potentially more commonly used) approaches for determining the number of dimensions, we wished to stay faithful to the approach used in Kwako et. al. (2019), who used the three fit-criteria (RMSEA, CFI and TLI) as guides for determining feasible factor models. We have added this methodological guidance to the beginning of Section 2.4.

R2.6.  For CFA, it would also be worthwhile to present other model fits (1, 2, or 4 factor or bifactor model), perhaps indicating the extent of residuals based on modification indices.

Response: Thank you for this suggestion. We have added the fit characteristics for the 1-, 2- and 4-factor models in the results, and added the factor loadings for the 4-factor model (Table 4), as well as a description of the (minor) differences between the 3- and 4-factor models.

R2.7. I feel a problem is that self-report and cognitive outcomes are different types of data, and for example, some scales (e.g., visual digit span) are latent dimensions due to strong covariation, i.e., the ordering is primarily a methodological effect.

Response:  The assessments included in the NIDA PhAB were a mix of both self-report questionnaire and laboratory performance tasks deemed by the expert panel to be the most appropriate/accessible validated tools available to assess each of the respective neurofunctional domains identified by that working group (as germane to SUD).  However, your point is well taken, and we agree that it would be preferable to separately analyze self-report from cognitive/task-based measures in future larger-scale validation studies of this battery. 

R2.8.  Although the discourse is based on sound professional foundations, I believe that the resulting analysis and results need more care and deeper interpretation.  The data analysis does not seem convincing to me.

Response:  We have added further clarifications and deeper description of our exploratory and confirmatory analyses in the Methods and Results Sections of the manuscript.

Reviewer 3 Report

Comments and Suggestions for Authors

This is a worthwhile study revealin emergent subtypes by an addiction-focused phenotpying battery. The topic of the study is very important and the results are very interesting. The paper has been well designed and conducted. I have no major comments to raise; only one minor suggestion:

It should extend the Discussion section with more clinical relevance / importance of the results.

Author Response

This is a worthwhile study revealing emergent subtypes by an addiction-focused phenotyping battery. The topic of the study is very important and the results are very interesting. The paper has been well designed and conducted. I have no major comments to raise; only one minor suggestion: It should extend the Discussion section with more clinical relevance / importance of the results.

Response: Thank you for your positive feedback.  The concluding paragraph has been slightly expanded to highlight potential implications. 

Reviewer 4 Report

Comments and Suggestions for Authors

In many ways, this is an excellent paper that reflects a major change in the NIDA/NIAAA approach to identifying antecedent cognitive and behavioral characteristics that are predictive of substance use disorder (SUD). It unquestionably  asserts the RDoC approach with confidence and attempts to validate some of the Redoc assumptions with actual data obtained from individuals who do and have never (I assume?) suffered from one or more of a limited number of substance abuse disorders. I frankly doubt that this approach will greatly inform the field about the causes of substance abuse or the most effective ways of treating them. But, it is an approach that is being strongly endorsed at present by major stock holders in the substance abuse endeavor and I believe that this paper reflects a major effort in that direction. It is clear, very well written and careful not to go overboard about the significance of its findings. 

Author Response

In many ways, this is an excellent paper that reflects a major change in the NIDA/NIAAA approach to identifying antecedent cognitive and behavioral characteristics that are predictive of substance use disorder (SUD). It unquestionably  asserts the RDoC approach with confidence and attempts to validate some of the Rdoc assumptions with actual data obtained from individuals who do and have never (I assume?) suffered from one or more of a limited number of substance abuse disorders. I frankly doubt that this approach will greatly inform the field about the causes of substance abuse or the most effective ways of treating them. But, it is an approach that is being strongly endorsed at present by major stock holders in the substance abuse endeavor and I believe that this paper reflects a major effort in that direction. It is clear, very well written and careful not to go overboard about the significance of its findings. 

Response: We thank you for your kind assessment of the study, and its current relevance to the field.

Round 2

Reviewer 2 Report

Comments and Suggestions for Authors

I appreciate the corrections and congratulate you on completing the paper.